# Does an Extraoral Suction Device Reduce Aerosol Generation and Prevent Droplet Exposure to the Examiner during Esophagogastroduodenoscopy?

**DOI:** 10.3390/jcm12072574

**Published:** 2023-03-29

**Authors:** Shintaro Fujihara, Hideki Kobara, Noriko Nishiyama, Naoya Tada, Yasuhiro Goda, Kazuhiro Kozuka, Takanori Matsui, Taiga Chiyo, Nobuya Kobayashi, Tatsuo Yachida, Tsutomu Masaki

**Affiliations:** 1Department of Gastroenterology and Neurology, Faculty of Medicine, Kagawa University, Kagawa 761-0793, Japan; 2Department of Gastroenterology, Kagawa Prefectural Shirotori Hospital, Kagawa University, Kagawa 769-2788, Japan

**Keywords:** esophagogastroduodenoscopy, aerosol, particle counter, COVID-19, adenosine triphosphate hygiene monitoring test

## Abstract

Esophagogastroduodenoscopy (EGD) is an aerosol-generating procedure. A major challenge in the COVID-19 era is how to prevent the spread of aerosols and droplets in endoscopic units. We evaluated the effectiveness of an extraoral suction device in preventing indoor aerosol diffusion and droplet exposure for examiners. The study involved 61 patients who underwent EGD at our institution from 1 February to 31 March 2022. To determine whether aerosol spread increases before or after EGD examination with an extraoral suction device located in front of the patient’s mouth, aerosols of 0.3, 0.5, 1, 3, 5, and 10 μm were measured with a handheld particle counter. The degree of contamination of the plastic gowns on the examiners was assessed using the rapid adenosine triphosphate test. The extraoral suction device significantly reduced the diffusion of large particles (3, 5, and 10 μm) after finishing the EGD examination. However, the diffusion of small particles (0.3 and 0.5 μm) was significantly increased. This extraoral suction device was effective in reducing large particle diffusion during EGD examination but was limited for minimizing small particle diffusion or droplet exposure to the examiner.

## 1. Introduction

The ongoing worldwide spread of severe acute respiratory syndrome coronavirus 2 (SARS-CoV-2), which caused the coronavirus disease (COVID-19) pandemic in 2019, has led to much debate regarding the predominant routes of transmission of SARS-CoV-2 [1]. SARS-CoV-2 can be transmitted from human to human and poses a higher risk in healthcare workers among the general population [2]. Additionally, the transmission of SARS-CoV-2 through aerosolization may be caused by aerosol-generating medical procedures including intubation, extubation, noninvasive ventilation, bronchoscopy, and manual ventilation in health care settings [3,4]. Esophagogastroduodenoscopy (EGD) is classified as an aerosol-generating procedure [5] and carries the risk of droplet exposure from patients [4]. Risk factors for aerosol generation include endoscopy-evoked coughing [6], burping [7], and a high body mass index [7]. Chan et al. [8] also reported a significant increase in particles of all sizes during upper gastrointestinal endoscopy.

EGD-evoked coughing is common. Thus, EGD in patients with a high risk of infection by SARS-CoV-2 or other respiratory pathogens should be performed with airborne personal protective equipment (PPE) and appropriate precautions [9]. It is necessary to develop equipment to prevent the spread of aerosols from EGD-evoked coughing and burping.

Novel protective equipment to reduce the spread of aerosols from the patient to the staff during endoscopic procedures has been proposed in various reports, including a specially designed acrylic box [7], face shield [10], and plastic sheet [11,12]. Although such protection systems prevent aerosol spread and droplet exposure to examiners, they do not have the ability to actively reduce aerosol production.

External suction devices, such as dental suction devices, are expected to actively protect the examiner from exposure to aerosols and droplets. Previous studies have shown that the application of dental aspirators may reduce the number of particles counted during endoscopy [8]. Continuous suctioning of the oral cavity contributes to aerosol reduction for the following reasons: (i) it reduces saliva retention and thus aspiration and patient coughing, and (ii) the aerosol generated during the procedure is partially aspirated. However, continuous suctioning produces physical stress and discomfort to the patients. Notably, dental suction devices have not been evaluated in terms of their ability to prevent droplet exposure in clinical practice. To resolve this issue, we focused on continuous extraoral aspiration using an extraoral suction device.

The purpose of this study was to verify whether an extraoral suction device used during EGD protects examiners from aerosol and droplet exposure using a particle counter and an adenosine triphosphate (ATP) contamination tester.

## 2. Materials and Methods

### 2.1. Patients

This retrospective observational study involved 61 patients undergoing diagnostic EGD in Kagawa Prefectural Shirotori Hospital from 1 February to 31 March 2022. The inclusion criterion was an age of >18 years. The exclusion criteria were emergency endoscopic treatment, such as that required for gastrointestinal bleeding, and unsuitability for the study as determined by the doctor in charge. Patient selection and group allocation are shown in Appendix A. All procedures were performed by two endoscopists (S.F. and Y.G.).

### 2.2. Extraoral Suction Device

For routine EGD, an extraoral suction device (Free100 Next^®^; Forest-one Corporation, Tokyo, Japan) was placed in front of the patient’s mouth (Figure 1). A large O-shaped hood was used as the tip attachment, and the power mode was set to vacuum level 5. The patient was placed in the left lateral position, and the endoscopist was positioned directly opposite the patient’s face.

### 2.3. Particle Measurement

A handheld optical particle counter (model 3889; Kanomax Japan Inc., Osaka, Japan) was used. The particle counter samples air at 2.83 L/min, detects particles by laser optical scattering, and reports particle number concentrations and size distributions in the range of 0.3 to 10 µm in diameter. The particle counter was calibrated at least 10 min before the first endoscopic procedure to ensure stable baseline readings.

All healthcare workers wore PPE for contact and droplet protection. Staff numbers and movement in rooms were kept to a minimum during the study period to minimize aerosol generation due to external factors. One fan coil unit was installed in the ceiling of the endoscopy unit, and regular air changes were performed. Endoscope cleaning systems were located in a separate room, and endoscopes were cleaned after each examination.

Once the patient entered the room, the particle counter was placed within 10 cm of the patient’s mouth, and measurements were taken at least 1 min before the procedure began. The measurement was continued during the procedure until after the patient left the endoscopy suite. Conductive silicone sampling tubing (of 1.5 m in length and 2 mm in internal diameter) was connected to the optical particle counter. Six particle sizes (0.3, 0.5, 1, 3, 5, and 10 µm) of the patients undergoing endoscopy were measured 60 s after the enclosure was installed, 60 s later (before endoscopy), continuously every 60 s during endoscopy, and 60 s after the endoscopy was completed.

### 2.4. ATP Measurement

Potential ATP contamination of the surface of the plastic gown was measured with LuciPac^®^ Pen and Lumitester PD-20^®^ System (Kikkoman Biochemifa Co., Tokyo, Japan) before and after the EGD. These devices detect concentrations that are expressed as the number of relative light units (RLU). Disposable plastic gowns were used during the EGD procedure. We then applied the same swab technique for all test area gowns by continuous painting from the top to the bottom in the examined area. The cutoff value to determine the degree of contamination by the rapid ATP test was set at >150 RLU.

### 2.5. Statistical Analyses

The primary outcome of the study was to compare aerosol generation during upper gastrointestinal endoscopy with baseline values immediately prior to the examination. The secondary outcomes of the study were the temporal dynamics of particle scattering and the presence of operator contamination. ATP values on the gown surface before and after EGD examination were compared. The Wilcoxon matched pairs signed-rank test was used to assess statistical differences between the primary outcome and ATP values. After the EGD was finished, the examiner’s ATP level of 150 RLU or more was considered to be contaminated. Data are expressed as mean ± standard deviation. The nonparametric Wilcoxon/Mann–Whitney U-test was used to examine statistical significance between the two groups. Continuous variables were compared using a Wilcoxon rank sum test. Categorical variables were compared using Fisher’s exact test when required. A *p* value of <0.05 was considered significant. All statistical analyses were performed using the Prism 6 software (GraphPad Software, La Jolla, CA, USA).

## 3. Results

The baseline characteristics of the 61 patients are shown in Table 1. Their mean age was 65.1 ± 16.1 years, and 36 were male. Of the 61 patients, transnasal endoscopy (TNE) was performed in 40 (65.6%) (40/61 cases) and transoral gastroscopy (TOG) was performed in 21 (34.4%). The mean procedure time was 188.4 ± 62.5 min, and biopsies were performed in 8 (13.1%) of the 61 patients. During the endoscopic examination, 19 (31.1%) patients exhibited burping, 12 (19.7%) exhibited vomiting, and 17 (27.9%) showed coughing.

The extraoral suction device significantly reduced the generation of large particles (3, 5, and 10 μm) after finishing the EGD examination. However, the generation of small particles (0.3 and 0.5 μm) was significantly increased (Figure 2, and Table 2).

The temporal changes in the particles during the EGD examination using the electric suction device are shown in Figure 3. Because of faulty measurement equipment during the examination, 7 of the 61 participants were unable to perform adequate testing with the particle counter. These seven patients were excluded from the analysis because of their short examination time, technical problems such as battery failure, or interruptions due to instrument malfunction. Therefore, 54 selective EGDs during which particles were continuously measured every minute during the examination were sampled. When six types of particles (0.3, 0.5, 1, 3, 5, and 10 μm) were analyzed before and after endoscopy in the same patients, the generation of small particles (0.3, 0.5, and 1.0 µm) increased during the examination. For larger particles (3, 5, and 10 μm), there was no increase during the first 3 min of the EGD examination, but an increase was observed after finishing the examination compared with the control (Appendix A).

The ATP levels before and after the test were 7.7 ± 6.4 and 71.4 ± 247.4 RLU, respectively (Figure 4, and Table 2). In addition, the degree of contamination of the examiner’s PPE was evaluated in 46 cases using ATP Lumitester, and contamination (RLU of >150) was found in 4 (8.7%) of these 46 cases.

A total of 61 patients were divided into two groups: nasal endoscopy (n = 40) and oral endoscopy (n = 21). Patient background factors such as age (*p* = 0.0197), mean procedure time (*p* = 0.0184) and burping (*p* = 0.0184) were significantly different between the two groups (Appendix A). Comparisons between the TNE and TOG groups showed no significant differences in the diffusion of aerosol of various sizes and ATP levels (Appendix A).

Therefore, the extraoral suction device suppressed the generation of large particles, but not that of small particles. Furthermore, the extraoral suction device did not prevent contamination of the examiner. No complications occurred in any patients during the current study.

## 4. Discussion

Extraoral suction devices were originally an effective means of reducing droplet dispersal during dental treatment and are useful for reducing the risk of spreading droplets during dental treatment [13]. Although the reduction in the risk of spreading droplets during dental treatment with extraoral suction devices has been studied, it has not yet been fully evaluated in the endoscopy field. This study produced three important clinical findings. First, the extraoral suction device exerted protection against exposure of the examiner to larger particle sizes (3, 5, and 10 μm). Second, the device exhibited a positive aspiration effect on larger particles under examination. Third, the protective effect of the device against droplet exposure to the examiner was limited. Respiratory droplets are usually divided into two size bins, large droplets (>5 μm in diameter) that fall rapidly to the ground and are thus transmitted only over short distances, and small droplets (≤5 μm in diameter) [14]. Extraoral suction devices actively aspirate the large droplets during the examination, thereby preventing the spread of respiratory droplet infection in the surrounding area [15]. On the other hand, they may be ineffective with regard to airborne transmission caused by small droplets (less than 5 µm in diameter).

Two recent studies provided evidence of aerosol generation during EGD examination using portable particle counters. Chan et al. [8] reported that aerosols were generated during EGD examination and that continuous aspiration with a dental suction device reduced the number of aerosols of all sizes. Sagami et al. [7] found that wrapping the patient’s head with plastic during EGD examination significantly increased the number of aerosols compared with a control group. TOG generated 1.96 times (*p* < 0.001) the number of background particles and TNE 2.00 times (*p* < 0.001) the number of background particles, while direct comparison showed that TOG generated 2.00 times more particles than TNE [16]. In the current study, we evaluated the changes in six different particle sizes before and after the EGD examination. The temporal changes in the three larger particles (3, 5, and 10 µm) decreased up to 3 min after the examination. However, the smaller particles (0.3 and 0.5 µm) showed an increase in number after finishing the EGD examination. Sunakawa et al. [15] showed that the maximum rate of increase tended to be higher for larger particle sizes (2 and 5 µm) in EGD examinations. The time of increase in the particle number due to the endoscopic procedure, as well as the maximum rate of increase, suggested that particles were generated not only in the patient’s mouth but also in the forceps’ holes. Therefore, the extraoral suction device did not reduce the maximum rate of increase or the time of increase in the particle count. In this study, the particle counter was placed at a distance of approximately 10 cm from the patient’s mouth, whereas in the study by Sunakawa et al. [15], the particle counter was placed 15 cm from the patient; this difference may have affected the test results. Placing the extraoral suction device near the patient’s mouth has the potential to reduce the number of some of the larger particles.

Particle counting alone cannot be used to directly evaluate the presence of viable viral material in droplets and aerosols. We therefore quantitatively assessed the droplet exposure to the examiner using ATP Lumitester. Sunakawa et al. [15] reported that an extraoral suction device increased the ATP levels in the patient and the examiner, which is consistent with the present results. However, they also reported reduced droplet contamination at a location further away from the patient when using the extraoral suction device compared with when not using the device. Furthermore, contamination was found within 1.5 m of the patient when an extraoral suction device was used; therefore, cleaning within 3 m of the examination site is important [15]. To date, the distance from the patient [17] and use of the prone position [18] have been effective in reducing droplet exposure during examination. The use of appropriate PPE is essential with an extraoral suction device.

Extraoral suction devices are mainly used in the dental field, and one of their disadvantages is noise as the vacuum airflow increases during the examination [19]. Ambient noise (measured as equivalent continuous sound pressure on a smartphone app) increased from 51 decibels during conversation to 72 dB when using vacuum level 5 and to 80 dB when using vacuum level 9. Noise of 70–80 dB is about as loud as a washing machine or dishwasher and does not affect the endoscopist’s performance. This increase in dB levels may affect healthcare provider communication, patient comfort, and staff safety monitoring [19].

This study had several limitations. First, it was a single-center study with a small number of patients. Second, we did not assess the relationship between the results of the ATP hygiene monitoring tests and microbiological tests. Third, the present study was not conducted in a clean or ultraclean room. Previous studies conducted in ultraclean rooms have allowed the sensitive detection of particle movement [6]. Particle counters are also affected by human movement and air conditioning; thus, in some clinical cases, it has been difficult to measure contamination in the surrounding environment. Unfortunately, only a small amount of literature has reported expected data on droplet dispersion during and after standard EGD examination [7,8]. Furthermore, no literature data on the burden of SARS-CoV-2 diffusion risks during gastrointestinal endoscopy have been reported. Because the results of this study may have been influenced by environmental factors in non-clean endoscopy rooms, the results should be studied under clean-room conditions in the near future. Finally, this study did not compare its results to those of a standard EGD examination without an extraoral suction device. Therefore, the contamination of plastic gowns was not compared with the contamination of gowns during procedures performed without a suction device, and contamination around the examination table was not considered.

## 5. Conclusions

The extraoral suction device used in this study was effective in reducing the diffusion of larger particles. Because its effect on reducing droplet exposure to the examiner is limited, endoscopy in combination with PPE is recommended.

## Figures and Tables

**Figure 1 jcm-12-02574-f001:**
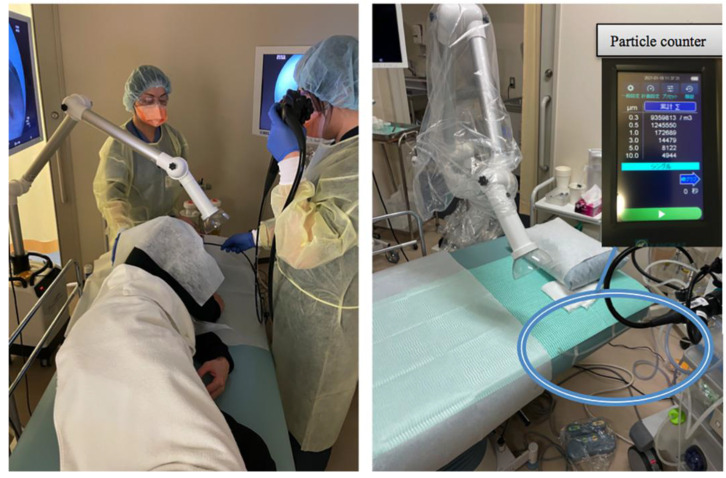
Aerosol particles were measured with a particle counter (model 3889; Kanomax Japan Inc., Osaka, Japan) in the patient’s mouth.

**Figure 2 jcm-12-02574-f002:**
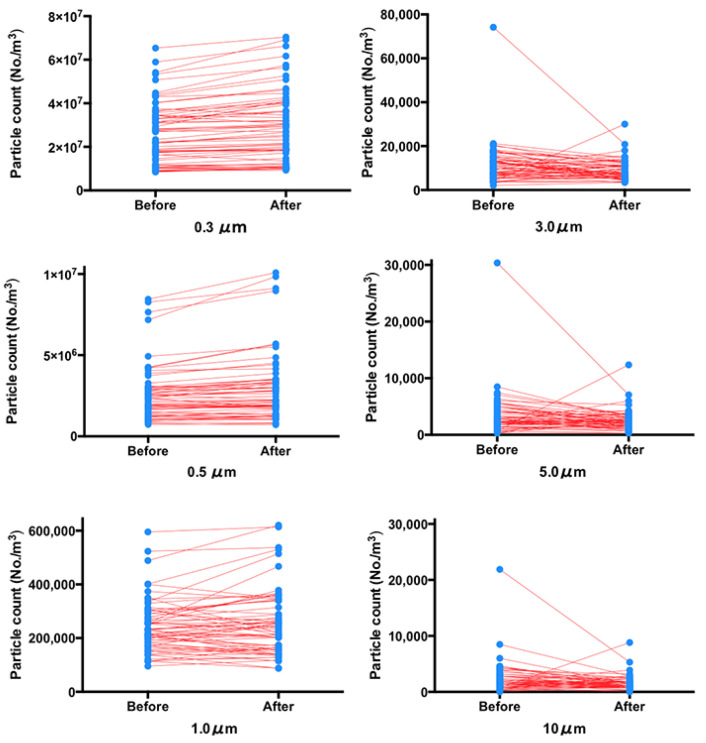
Changes in the six particles before and after the esophagogastroduodenoscopy examination using the extraoral suction device (n = 61).

**Figure 3 jcm-12-02574-f003:**
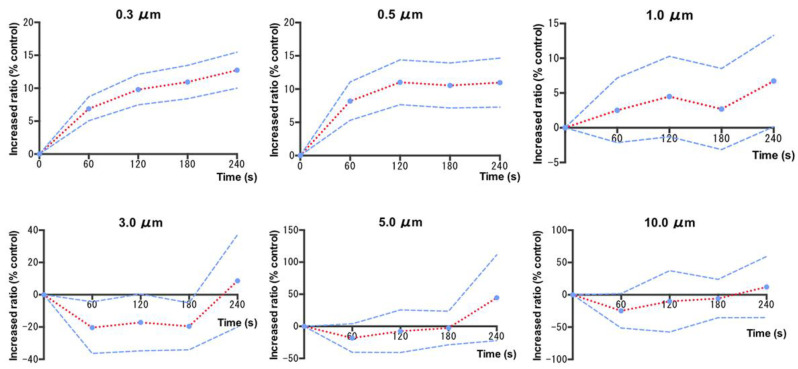
Percentage increase in aerosol particles of 0.3, 0.5, 1, 3, 5, and 10 µm in size at 60, 120, 180 and 240 s from the start of esophagogastroduodenoscopy (n = 54). The lines represent the mean particle count (0.3, 0.5, 1, 3, 5, and 10 μm) of the 54 patients using the extraoral suction device. The blue dotted line indicates the 95% confidence interval.

**Figure 4 jcm-12-02574-f004:**
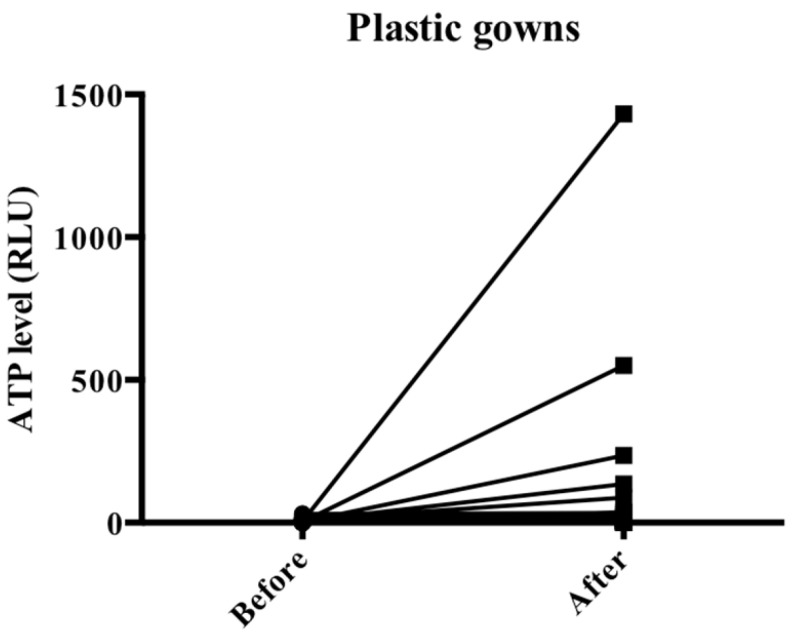
Changes in ATP concentrations with disposable plastic gowns. ATP, adenosine triphosphate.

**Table 1 jcm-12-02574-t001:** Patient characteristics.

	n = 61
Mean age ± SD, y	65.1 ± 16.1
Male sex, no (%)	36 (59.0)
Medical history, n (%)	21 (34.4)
Nasal endoscope, n (%)	40 (65.6)
Mean procedure time ± SD, s	188.4 ± 62.5
Cough, n (%)	17 (27.9)
Reflex vomiting, n (%)	12(19.7)
Sneezing, n (%)	0 (0)
Burping, n (%)	19 (31.1)
Body movement, n (%)	3 (4.9)
Interventions, n (%)	
No	53 (86.9%)
Biopsy	8 (13.1)

SD, standard deviation.

**Table 2 jcm-12-02574-t002:** Comparison of changes in aerosol counts and ATP levels before and after esophagogastroduodenoscopy.

	Before	After	*p*-Value
Mean counts (×10^6^/m^3^) ± SD			
0.3 μm	25.9 ± 13.7	28.9 ± 16.1	<0.0001
0.5 μm	2.6 ± 1.7	2.9 ± 2.1	<0.0001
1.0 μm	0.2 ± 0.1	0.3 ± 1.2	0.324
Mean counts (×10^3^/m^3^) ± SD			
3.0 μm	11.5 ± 9.3	8.9 ± 4.7	0.003
5.0 μm	3.3 ± 3.9	2.5 ± 1.8	0.019
10.0 μm	2.3 ± 2.9	1.5 ± 1.3	0.0009
ATP level (RLU)	7.7 ± 6.4	71.4 ± 247.4	0.0103

ATP, adenosine triphosphate; SD, standard deviation; RLU, relative light units.

## Data Availability

Not appliable.

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
