# Peer review of "Does an Extraoral Suction Device Reduce Aerosol Generation and Prevent Droplet Exposure to the Examiner during Esophagogastroduodenoscopy?"

_jcm, 2023, doi:10.3390/jcm12072574_

Round 1

Reviewer 1 Report

I think the subject of the study is very interesting. However, there are many things to correct. The authors acknowledge several limitations including the low number of patients analyzed. But in addition to being a low number of procedures performed in a single center there is an important point is that two different endoscopic techniques were used. Transnoral endoscopy probably differs from transnasal endoscopy in cough production and the diameter of the endoscope is different. In this sense, the analysis of each technique should be carried out separately (which would further diminish the validity given the low number of procedures of each technique). On the other hand, they must make a better description of the endoscopic protocol: cleaning system performed prior to endoscopy; existence or not of any air circulation system within the room, number of people (in addition to the patient) included in the endoscopic room, use or non-use of face mask and type of marking used by the staff, etc ... On the other hand in the discussion (especially with the findings of different change in quantity of small particles vs larger particles) It is important to discuss if there are differences in transmission capacity of microorganisms and types of microorganisms according to the size of these particles. Finally, I suggest correcting figure 1 since the text boxes cover an important part of the image which decreases the ability to observe the details of the technique used for extraoral aspiration and particle measurement.

Author Response

Reviewer 1

  1. I think the subject of the study is very interesting. However, there are many things to correct. The authors acknowledge several limitations including the low number of patients analyzed. But in addition to being a low number of procedures performed in a single center there is an important point is that two different endoscopic techniques were used. Transnoral endoscopy probably differs from transnasal endoscopy in cough production and the diameter of the endoscope is different. In this sense, the analysis of each technique should be carried out separately (which would further diminish the validity given the low number of procedures of each technique).

Response: Thank you for pointing this out. A similar point was raised by reviewer 2, so we have added a supplementary table 1 and table 2 for the data comparing transnasal endoscopy (TNE) and transoral endoscopy (TOG). Comparisons between TNE and TOG groups showed no significant differences in aerosol of various sizes and ATP levels. This is due to EGD-evoking cough has been reported as a factor that increases aerosols and no differences were found in the two groups In addition, some content has been added in the results for the relevant sections.

  1. On the other hand, they must make a better description of the endoscopic protocol: cleaning system performed prior to endoscopy; existence or not of any air circulation system within the room, number of people (in addition to the patient) included in the endoscopic room, use or non-use of face mask and type of marking used by the staff, etc ...

Response: Thank you for pointing this out. The factors you pointed out are very important for the interpretation of the results. All healthcare workers wore PPE (including face masks) for contact and droplet protection. Staff numbers and movement in rooms were kept to a minimum during the study period to minimise aerosol generation due to external factors. One fan coil unit was installed in the ceiling of the endoscopy unit and regular air changes were performed. Endoscope cleaning systems were located in a separate room and endoscopes were cleaned after each examination.

We have therefore made additions and corrections to the methods section.

  1. On the other hand in the discussion (especially with the findings of different change in quantity of small particles vs larger particles) It is important to discuss if there are differences in transmission capacity of microorganisms and types of microorganisms according to the size of these particles.

Response: Thank you for pointing this out. The size of the COVID-19 virus was 0.06–0.14 μm, with nanospikes coated on its spherical viral envelope with heights of 0.09–0.12 μm attached to a larger carrier and becoming airborne with a size of 0.1–0.3 μm. Respiratory droplets are usually divided into two size bins, large droplets (>5 μm in diameter) that fall rapidly to the ground and are thus transmitted only over short distances, and small droplets (≤5 μm in diameter). Therefore, extraoral suction device actively aspirate the large droplets during the examination, thereby preventing the spread of respiratory droplet infection in the surrounding area. On the other hand, they may be ineffective with regard to airborne transmission caused by small droplets (less than 5 µm in diameter). Reviewer 2 made a similar point, so we made some changes to the discussion section and added reference 19.

  1. Finally, I suggest correcting figure 1 since the text boxes cover an important part of the image which decreases the ability to observe the details of the technique used for extraoral aspiration and particle measurement.

Response: Thank you for your valuable suggestions, we have made the corrections from Figure 1, except for the text box.

Reviewer 2 Report

Does this machine have potential to prevent endoscopists from viral infection during EGD, anyway? The device can vacuum large aerosols, but not small ones. What does this mean in infection control during gastrointestinal endoscopy? 

How noisy is this machine (I can not imagine 72dB)? Can this level of noise influence endoscopists’ performance?

Do the authors currently use this machine during EDG? If no, I would like to ask reasons.

In this study, 65% of the participants had nasal endoscopy. It seems that cough is less serious during nasal EGD than oral EGD. This might influence the results of this study. I am interested in comparison between the nasal and oral group in terms of aerosol contamination. 

Author Response

Reviewer 2

  1. Does this machine have potential to prevent endoscopists from viral infection during EGD, anyway? The device can vacuum large aerosols, but not small ones. What does this mean in infection control during gastrointestinal endoscopy? 

Response: Thank you for pointing this out.

Thank you for pointing this out. The size of the COVID-19 virus was 0.06–0.14 μm, with nanospikes coated on its spherical viral envelope with heights of 0.09–0.12 μm attached to a larger carrier and becoming airborne with a size of 0.1–0.3 μm. Respiratory droplets are usually divided into two size bins, large droplets (>5 μm in diameter) that fall rapidly to the ground and are thus transmitted only over short distances, and small droplets (≤5 μm in diameter). Therefore, extraoral suction device actively aspirate the large droplets during the examination, thereby preventing the spread of respiratory droplet infection in the surrounding area. On the other hand, they may be ineffective with regard to airborne transmission caused by small droplets (less than 5 µm in diameter). Reviewer 1 made a similar point, so we made some changes to the discussion section and added reference 19.

  1. How noisy is this machine (I can not imagine 72dB)? Can this level of noise influence endoscopists’ performance?

Response: Thank you for your valuable remarks. 70 decibels is about as loud as a washing machine or dishwasher and does not affect the endoscopist's performance.

The use of the machine did not significantly impede communication with the patient during the examination, but the examiner had to speak somewhat louder during the conversation. We have added a Discussion section on noise levels.

  1. Do the authors currently use this machine during EDG? If no, I would like to ask reasons.

Response: Recently, an assessment of air contamination by SARS-CoV-2 in hospitals has shown that air near and far from contaminated patients can carry viral RNA (1). In addition, it has been suggested that aerosol transmission is increased with the newer types (alpha and delta), emphasizing the need to prevent aerosolization in hospitals, especially in endoscopy laboratories emphasizes the need to prevent aerosolization by all means (2).

Extraoral suction devices do not inhibit the generation of small droplets, which is a problem in aerosolisation. We have previously developed a patient-covering negative-pressure box to control small droplets and droplet exposure (3), which is now mainly used in endoscopy unit.

  1. Birgand G, Peiffer-Smadja N, Fournier S et al. Assessment of air contamination by SARS-CoV-2 in hospital settings. JAMA Netw Open. 2020;3:e2033232.
  2. Port J R, Yinda C K, Avanzato V A et al. Increased small particle aerosol transmission of B.1.1.7 compared with SARS-CoV-2 lineage A in vivo. Nat Microbiol. 2022;7:213–223.
  3. Fujihara S, Kobara H, Nishiyama N, Tada N, Kozuka K, Matsui T, Chiyo T, Kobayashi N, Shi T, Yachida T, Uchida T, Nagatomi T, Oba H, Masaki T. Clinical Efficacy of Novel Patient-Covering Negative-Pressure Box for Shielding Virus Transmission during Esophagogastroduodenoscopy: A Prospective Observational Study. Diagnostics (Basel), 2021, 11, 1679. 

  1. In this study, 65% of the participants had nasal endoscopy. It seems that cough is less serious during nasal EGD than oral EGD. This might influence the results of this study. I am interested in comparison between the nasal and oral group in terms of aerosol contamination. 

Thank you for pointing this out. A similar point was raised by reviewer 2, so we have added a supplementary table 1 and table 2 for the data comparing transnasal endoscopy (TNE) and transoral endoscopy (TOG). Comparisons between TNE and TOG groups showed no significant differences in aerosol of various sizes and ATP levels (Supplemental table 2). This is due to EGD-evoking cough has been reported as a factor that increases aerosols and no differences were found in the two groups In addition, some content has been added in the results for the relevant sections.

Round 2

Reviewer 1 Report

The new version is much better

Author Response

Thank you for your valuable comments regarding the limitations of this study. Also, thank you very much for taking time out of your very busy schedule to check every detail of our paper and for your appropriate comments.

Reviewer 2 Report

The revised manuscript has been improved. 

Author Response

(The authors gave the same response as above.)
